# Assessing the Performance of the Satellite-Based Precipitation Products (SPP) in the Data-Sparse Himalayan Terrain

Sonu Kumar [1,2], Giriraj Amarnath [3], Surajit Ghosh [3], Edward Park [1,2,*], Triambak Baghel [4], Jingyu Wang [1], Malay Pramanik [5] and Devesh Belbase [4]

1. National Institute of Education (NIE), Nanyang Technological University (NTU), 1 Nanyang Walk, Singapore 637616, Singapore
2. Earth Observatory of Singapore (EOS), Nanyang Technological University (NTU), 50 Nanyang Avenue, Singapore 639798, Singapore
3. International Water Management Institute (IWMI), 127, Sunil Mawatha, Pelawatte, Battaramulla, Colombo 2075, Sri Lanka
4. Water Engineering and Management, Asian Institute of Technology, Pathum Thani 12120, Thailand
5. Urban Innovation and Sustainability Program, Department of Development and Sustainability, Asian Institute of Technology, Pathum Thani 12120, Thailand
* Correspondence: edward.park@nie.edu.sg

**Abstract:** Located on the south-facing slope of the Himalayas, Nepal receives intense, long-lasting precipitation during the Asian summer monsoon, making Nepal one of the most susceptible countries to flood and landslide hazards in the region. However, sparse gauging and irregular measurement constrain the vulnerability assessments of floods and landslides, which rely highly on the accuracy of precipitation. Therefore, this study evaluates the performance of Satellite-based Precipitation Products (SPPs) in the Himalayas region by comparing different datasets and identifying the best alternative of gauge-based precipitation for hydro-meteorological applications. We compared eight SPPs using statistical metrics and then used the Multi-Criteria Decision-Making (MCDM) technique to rank them. Secondly, we assessed the hydrological utility of SPPs by simulating them through the GR4J hydrological model. We found a high POD (0.60–0.80) for all SPPs except CHIRPS and PERSIANN; however, a high CC (0.20–0.40) only for CHIRPS, IMERG_Final, and CMORPH. Based on MCDM, CMORPH and IMERG_Final rank first and second. While SPPs could not simulate daily discharge (NSE < 0.28), they performed better for monthly streamflow (NSE > 0.54). Overall, this study recommends CMORPH and IMERG_Final and improves the understanding of data quality to better manage hydrological disasters in the data-sparse Himalayas. This study framework can also be used in other Himalayan regions to systematically rank and identify the most suitable datasets for hydro-meteorological applications.

**Keywords:** satellite precipitation products; discharge simulation; hydrological modeling; Nepal

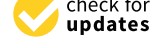



## 1. Introduction

The sparse distribution of meteorological stations in the Himalayan region coupled with the high-altitude terrain, poses a considerable challenge in investigating hydrometeorological events with precision [1–3]. The rain gauge network is the most trusted instrument for precise precipitation measurement of ground-based observations [4,5]. In remote areas (e.g., mountainous terrain), sparse and irregular distribution of rain gauges, limited spatial representativeness of local measurements, and damage to climatological stations from natural disasters are some of the most common causes of inaccurate precipitation [6,7]. At present, a wide range of Satellite Precipitation Products (SPPs) and Reanalysis Precipitation Products (RPPs) are available, which could provide viable alternatives for precipitation datasets in the data-sparse region. Aided with innovation in satellite techniques and retrieval algorithms, many SPPs have been developed globally with increasing

temporal and spatial resolutions. However, knowledge of their applicability for various studies, e.g., hydropower assessment, hydrological modeling, flood and drought, climatic changes, and understanding the precipitation characteristics over complex mountainous terrain, is still limited because of the spatial heterogeneity in their performance [8–13]. Several studies have recognized that the performance of SPPs and RPPs in terms of detecting the rainfall occurrence (yes/no) and magnitude of rainfall is greatly influenced by several factors, e.g., elevations, spatial and temporal resolutions, and selected performance indicators [14–16]. Therefore, it is recommended to evaluate the SPPs and RPPs before their application for any hydro-meteorological hazards assessment.

Many studies developed frameworks to evaluate the SPPs [17,18] using several categorical and continuous variable performance indicators globally [19–22]. For example, most previous frameworks employed Probability of Detection (POD), False Alarm Ratio (FAR), Peirce Skill Score (PSS), and Critical Successive Index (CSI) as categorical variable indicators, and Root Mean Square Error (RMSE), Correction Coefficients (CC), Kling–Gupta Efficiency (KGE), and Percentage Bias (PBIAS) as continuous variable indicators in comparing the performance of SPPs and RPPs [16,17,23–25]. For example, Satgé et al. [17] assessed the reliability of 23 SPPs using KGE throughout West Africa. In Asia, Wang et al. [26] assessed the suitability of ERA-Interim, JRA-55, and NCAR-1 based on CC, PBIAS, RMSE, and MAE for the Tibetan Plateau in China and found all selected datasets to be performing differently based on the performance indicators employed. Dandridge et al. [18] evaluated two GPPs in the Lower Mekong River Basin across Southeast Asia using MAE, CC, RSME, and BIAS. Chawla et al. [8] concluded similarly while evaluating four RPPs, SPPs, and Weather Research and Forecasting (WRF) datasets for the Himalayan Region. Sharma et al. [15] analyzed the performance of only four datasets, IMERG_Early, IMERG_Final, GSMaP-MVK, and GSMaP-Gauge, using CC, and RMSE, with similar findings to Wang et al. [26] and Chawla et al. [8]. However, there are many disadvantages to using the above-discussed frameworks. First, these only evaluate SPPs against ground observed rainfall and might not be suitable for hydrological applications. These frameworks use various performance indicators that might puzzle SPPs users in selecting the best datasets since the performance of SPPs changes from one performance indicator to another [19–22].

Simultaneously, a few researchers developed more comprehensive and robust frameworks for comparing the performance of SPPs globally based on rainfall capturing and discharge simulation ability [27–29]. By employing similar approaches, several studies (e.g., Jiang et al. [30]; Chawla et al. [8]; Tan et al. [7]) attempted to investigate the hydrological suitability of SPPs and RPPs by using them as input to hydrological models. Unfortunately, these studies also observed the performance of SPPs varies based on selected performance indicators, e.g., TRMM outperformed in terms of POD and worst in terms of FAR simultaneously. As a whole, all previous studies determined that the performance of SPPs varies significantly with the selected performance indicators [6,25,31,32]. Although several performance indicators have been used for a few outdated versions of datasets, no single comprehensive indicator could be employed to evaluate the latest datasets extensively. In addition, no studies were found for assessing the suitability of the SPPs in the selected area because of their location. Thus, the lack of comprehensive assessment limits the appropriate selection of SPPs and RPPs for herpetological application, e.g., monitoring and early warning of floods and landslides and hydro-meteorological modeling.

Keeping in mind the same, this paper developed a novel framework that uses various performance indicators and Multi-Criteria Decision Making (MCDM) to comprehensively assess the latest versions of SPPs and offer suitable recommendations for hydro-meteorological studies. The framework developed (Figure 1) was employed in Simat Khola River Basin (SKRB), Nepal, as shown in Figure 2. SKRB represents the complex mountainous Himalayan region well. This case study employed three specific steps in addressing the goal, where (i) a total of eight SPPs were compared against ground-based precipitation observation using eight performance indicators; (ii) SPPs were ranked using the MCDM method; and (iii) the utility of the best-performing SPPs for hydrological application was as-

sessed using the GR4J hydrological model. The upper and lower part of the basin is highly vulnerable to recurrent instances of drought and floods, leading to a significant loss of life and property due to the geographical location and complex mountainous topography [33]. The basin is seated well within the mountainous region, with only a few rainfall stations available in the upper catchment; this hampers the hydro-meteorological understanding and timely early warnings during significant disasters. Although the study area is small, it has the benefit of improved accuracy because the routing processes will not dominate the rainfall-runoff process during hydrological simulation [25]; the basin thus qualifies as a case study area.

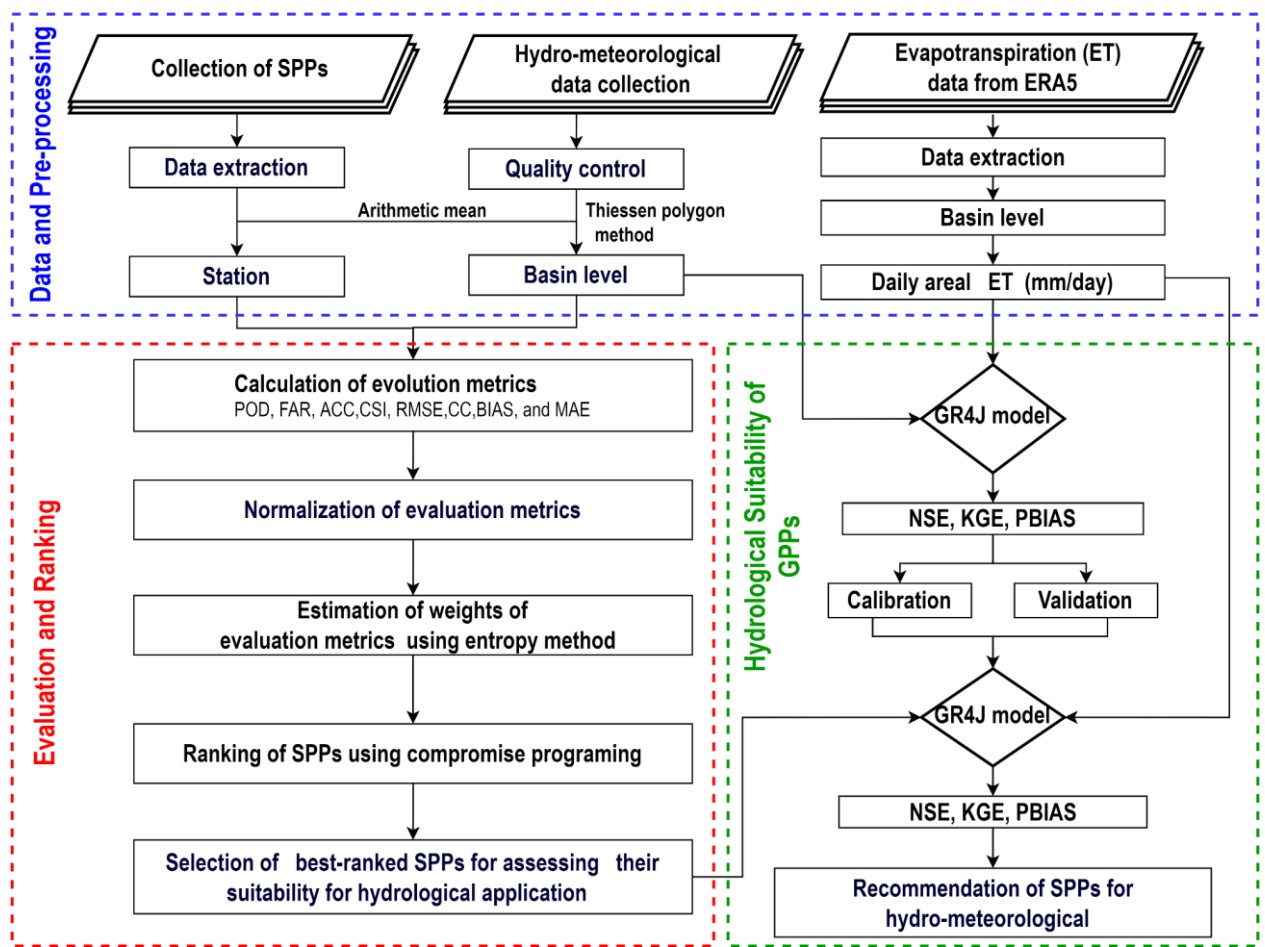

**Figure 1.** An overall developed methodological framework which comprehensively evaluates and ranks the SPPs and assesses their hydrological utility.

The framework developed for this study will significantly contribute to addressing various methodological challenges in selecting the best-suited datasets for different hydro-meteorological applications based on multiple performance indicators. Our framework has strong potential to be employed in different data-sparse high-altitude regions across the world to systematically rank and identify the most suitable datasets. This study contributes significantly to understanding the performance of the latest version of SPPs, which could aid in improved management of hydrological disasters for the study basin and the data-sparse highland terrain. It can also benefit the atmospheric and climate research community by enhancing the algorithm used to derive SPPs, which may improve their performance. The research findings might aid researchers and the local governments in Nepal and India in their understanding of the quality of SPPs—this, in turn, can allow them to select the appropriate SPPs for water-related policy making and monitoring and conducting research on various water-induced disasters, e.g., landslides, floods, and drought in the basin.

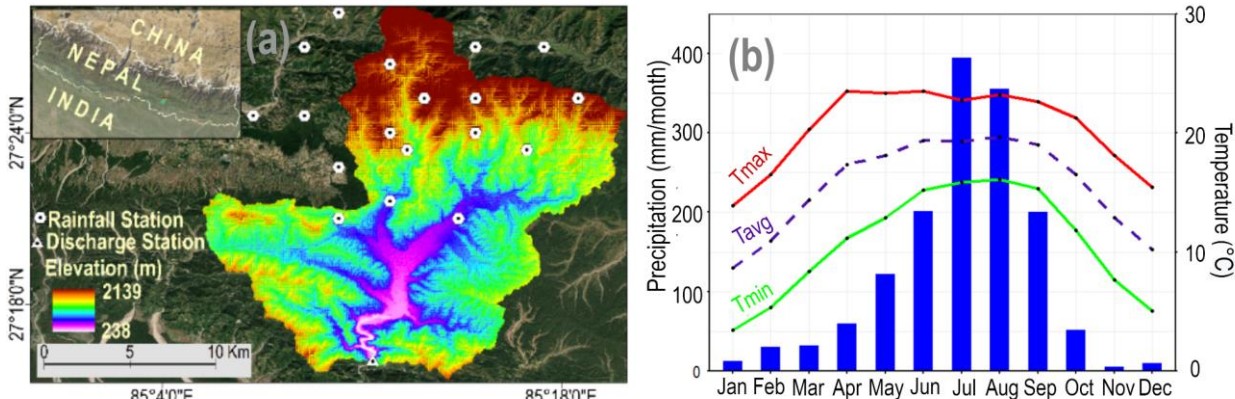

**Figure 2.** (**a**) Location of Simat Khola River Basin (SKRB), Nepal with physiography and selected ground-based to observed precipitation points; (**b**) climatograph of study basin for 16 years from 2003 to 2018. The bar diagram denotes the mean of seasonal monthly accumulated rainfall, and red solid, blue dotted, and solid green lines represent the monthly maximum, average, and minimum temperature, respectively.

## 2. Materials and Methods

### 2.1. Study Area

Nepal's Simat Khola River Basin (SKRB) was selected as a case study region to evaluate and choose the most suitable SPPs (Figure 2). SKRB lies between the eastern part of the Bagmati basin and the southwest part of the Gandaki basin. Its geographical coordinates are 85°40′E to 85°20′E longitude and 27°10′N to 27°35′N latitude, covering an area of 301 sq. km. The basin's topography varies in the range of 238–2139 m above mean sea level (MSL). SKRB's low land area (215–1000 m) has temperature varying in the range of 20–30 °C, the foothills (1000–2000 m) have a temperature variation of 15–20 °C, and the mountainous region (2000 m and above) has temperatures ranging 10–15 °C. The annual precipitation of the basin is 1756.8 mm, which is distributed in a nonhomogeneous manner. Approximately 77% of the rainfall occurs during the monsoon season (June–September), and about 15% occurs in the pre-monsoon season (March-May).

### 2.2. Observed Hydro-Meteorological Data

The gauge-based rainfall observation (2003–2018) for 18 rain gauge stations and discharge (2003–2014) was collected for the Khokana stream gauge station (station number 550) as an outlet of the basin from the Department of Hydrology and Meteorology (DHM), Nepal. DHM measured daily rainfall at 03 UTC using eight-inch diameter manual rain gauges [34]. All gauge-based observed rainfall with fewer missing values (less than 5%) within and near basins were chosen for evaluation (refer to Figure 2a). The widely accepted Thiessen Polygon (TP) method was employed to estimate the daily areal precipitation across the basin from point gauge-based observed rainfall. The period of missing value was excluded when comparing the gauge-based rainfall with SPPs to avoid any additional uncertainty due to interpolation. Before analysis, we aggregated all SPPs, excluding CHIRPS and PERSIANN_CSR, since these datasets are available at a daily time scale with the same time windows as DHM daily precipitation (03 UTC) to accumulate daily precipitation. This study employed daily observed discharge from 2003 to 2007 and 2010 to 2014 for hydrological model calibration and validation, respectively. Missing discharge was observed from 2006 to 2009; therefore, this period was excluded from the model development. Reanalysis-based Potential Evapotranspiration (RPET) from the ERA5 dataset was used as input, apart from precipitation (refer to Supplementary Text S1 and Table S1) for a brief description of the data used and its processing.

*2.3. Satellite-Based Precipitation Products (SPPs)*

A total of eight most widely used publicly available SPPs CHIRPS, CMORPH V02, IMERG Early V06, IMERG Late V06, IMERG Final V06 (hereafter referred to as IMERG_Early, IMERG_Late, IMERG_Final), PERSIANN, PERSIANN_CCS, and PERSIANN_CDR were selected (refer to Supplementary Text S2 for a brief description of data). Their characteristics, such as temporal and spatial resolutions, spatial and temporal coverage, and latency of datasets, have been described in Table 1. CHIRPS (Climate Hazards Group InfraRed Precipitation with Station data) delivers inclusive, consistent, reliable, up-to-date datasets for real-time application, including trend analysis and seasonal drought analysis. CHIRPS uses infrared Cold Cloud Duration (CCD) observations, satellite imagery, and gauge observed rainfall to estimate daily rainfall based on interpolation techniques (Table 1). Further technical details about CHIRPS have been available [35–39]. NOAA CPC Morphing Technique (CMORPH) is a very high spatial (8 km at the equator) and temporal (30 min) resolution precipitation estimate. It is derived from low orbiter satellite microwave observations, geostationary satellite I.R., and passive microwave data using the CPC Morphing technique [40]. Integrated Multi-Satellite Retrievals for GPM (IMERG) precipitation datasets are created by merging data from all GPM satellite constellations. IMERG provides three types of products, namely IMERG_Early, IMERG_Late, and IMERG_Final, for different users after observations of 4 h, 14 h, and 3.5 months (see Table 1). Evaluating IMERG products with different latency provides the baseline quantification of uncertainty when IMERG_Early, and IMERG_Late are applied to the near-real-time hazard monitoring and assessment. Precipitation Estimation from Remotely Sensed Information using Artificial Neural Networks (PERSIANN) uses the neural network approximation method to estimate the precipitation rate from infrared brightness temperature images provided by geostationary satellites (i.e., GOES-8, GOES-10, GMS-5, Metsat-6, and Metsat-7) [41,42]. PERSIANN-Cloud Classification System (PERSIANN_CCS) provides real-time precipitation datasets; it has been developed using the classification of cloud-patch features based on cloud height, areal extent, and variability of texture estimated from satellite imagery [41,43]. PERSIANN-Climate Data Record (PERSIANN_CDR) provides long-term precipitation data. As a consistent, long-term, high-resolution, and global precipitation dataset, it is useful for studying the changes and trends in daily precipitation (especially extreme precipitation events) due to climate change and natural variability [41,44].

*2.4. Statistical Evaluation Method*

A point-to-grid comparison was applied in this study, where the SPP data grid is compared to its closest gauge station. Given that one SPP grid box could contain multiple gauge stations, this grid box is compared multiple times against the stations within; therefore, the evaluation matrix is strongly affected by the homogeneity of the enclosed point-based observations. Notably, the oversampling in this point-to-grid comparison seems to magnify the sub-grid heterogeneity. However, this is exactly one of the characteristics of precipitation distribution over complex terrain, which should be taken into consideration. This method has been widely used in precipitation evaluation studies [47], where the gridded precipitation dataset usually contains multiple point-based observations, and each point is compared independently with the enveloping grid. Furthermore, this method noticeably avoids creating additional errors due to interpolation in observed data and makes the best use of the valuable rain gauge observations [31,48–50]. Therefore, the suitability of SPPs was assessed against each selected gauge-observed precipitation data on categorical and continuous variable statistical matrices [51].

This study employed two statistical approaches to compare the performance of SPPs. The categorical variable verification approaches use dichotomous verification, i.e., yes/no; here, 'yes' and 'no' indicate whether the event happened or not, respectively. This study used a threshold value to differentiate between events and not events as 1 mm/day to avoid human error while taking a reading at gauge stations [52]. The categorical variable verification of SPPs was performed using four widely used categorical variable performance indicators, namely the Potential of Detection (POD), False Alarm Ratio (FAR), Accuracy

(ACC), and Critical Successive Index (CSI). The second statistical approach utilized in the study is continuous variable verification which measures how the magnitude of SPPs estimates differs from gauge observations. Continuous variable performance indicators—Root Mean Square Error (RMSE), Pearson Correlation Coefficient (CC), and Mean Absolute Error (MAE)—were used to check the accuracy of a continuous variable. These well-known performance indicators have been briefly mentioned, along with their formula, in Text S3 of the supplementary document.

**Table 1.** A brief description of the datasets used in the present study.

| Dataset | Input Data | Method and Technique | Lowest Frequency | Spatial Resolution | Temporal Coverage | Latency | Reference |
|---|---|---|---|---|---|---|---|
| Terrain (Digital Elevation Model) | Radar interferometry | NASA Shuttle Radar Topographic Mission (SRTM) | - | 30 m × 30 m | - | - | https://lpdaac.usgs.gov/about/citing_lp_daac_and_data (accessed on 1 January 2021). |
| Gauge-based Precipitation | - | Gauge | h | 18 stations | 2003–2017 | 1y | http://www.dhm.gov.np/ (accessed on 1 January 2020). |
| Discharge | - | - | h | 1 station | 2003–2015 | NA | http://www.dhm.gov.np/ (accessed on 1 January 2020). |
| Reanalysis-based Potential Evapo-transpiration (RPET) | Model and gauge-based observations | 4D-Var data assimilation and model forecasts | h | 0.25° × 0.25° | 1979–Present | 5 d | [45] |
| CHIRPS | Infrared Cold Cloud Duration (CCD) observations, satellite imagery, and ground-based observed rainfall | Interpolation techniques | d | 0.05° × 0.05° | 1981-Present | 1 m | [35] |
| CMORPH V1.0 | Low orbiter satellite microwave observations and geostationary satellite IR data | CPC MORPHing technique | 30 min | 0.25° × 0.25° | 2002-Present | NA | [40] |
| IMERG_Early (06) | Temperature and humidity ancillary data from a constellation of passive microwave satellites | Forward propagation GPROF algorithm | 30 min | 0.1° × 0.1° | 2000-Present | 4 h | [46] |
| IMERG_Late (06) | Same as IMERG_Early | Both forward and backward propagation GPROF algorithm | 30 min | 0.1° × 0.1° | 2000-Present | 12 h | [46] |
| IMERG_Final (06) | Same as IMERG_Late | Same with IMERG_Late GPROF algorithm and Gauge correction technique | 30 min | 0.1° × 0.1° | 2000-Present | 3.5 m | [46] |
| PERSIANN | Geostationary longwave infrared imagery | Neural network function classification procedures | h | 0.25° × 0.25° | 2000-Present | 2 d | [41] |
| PERSIANN_CCS | Satellite imagery | Variable threshold cloud segmentation | h | 0.04° × 0.04° | 2003-Present | 1 d | [41] |
| PERSIANN_CDR | GridSat-B1 infrared data and GPCP data | Artificial Neural Networks—Climate Data Record | d | 0.25° × 0.25° | 1983-Present | NA | [41] |

Note: min, h, d, m, and NA denote the minute, hour/hourly, day/daily, month/monthly, and not available, respectively. This study evaluated all selected SPPs from 2003 to 2017.

### 2.5. Ranking of SPPs Using MCDM Method

This study employed compromise programming as MCDM to rank SPPs on the various performance indicators determined previously. Compromise programming assigned the ranks of SPPs based on Lp values—the minimum distance between normalized values and ideal values performance indicator [53,54]. Before this, the entropy method was used to calculate the weightage of each performance indicator; it assigned the weights of different indicators using equations (Equations (1) and (2)) [32,54]. After determining the entropy matrix, the degree of diversification was calculated (Equation (3)). Then, the weight of each

indicator was determined (Equation (4)). Finally, the distance of each normalized indicator from its normalized ideal values (*Lp*) was measured (Equation (5)) to assign the rank.

$$En_j = -\frac{1}{ln\ (T)} \sum_{a=1}^{T} k_{aj} ln ln\ (k_{aj})\ for\ j = 1,\ 2,\ \ldots\ldots J \tag{1}$$

$$k_{aj} = \frac{k_j}{\sum_{a=1}^{T} k_j(a)} \tag{2}$$

$$D_{dj} = 1 - En_j \tag{3}$$

$$W_j = \frac{D_{dj}}{\sum_{j=1}^{J} D_{dj}} \tag{4}$$

$$L_p = \left[ \sum_{j=1}^{J} w_j^p \left| f_j^* - f_j(a) \right|^p \right]^{\frac{1}{p}} \tag{5}$$

where $En_j$ denotes the entropy of represents every single indicator for $j$, $T$ represents the total number of global precipitation datasets, $k_j(a)$ denotes the value of chosen indicator $j$ for SPPs $a$, $Kaj$ represents the normalized value of error indices, and $J$ represents the maximum number of indicators. $Ddj$ represents the degree of diversification, and $Wj$ represents the normalized weight of indicators. $Lp(a) = Lp$ metric for SPPs a for the chosen value of parameter $p$; $f_j(a)$ = Normalized value of indicator $j$ for SPPs a; $f_j^*$ = Normalized ideal value of indicator $j$; $p$ = Parameter (1 for linear, 2 for squared Euclidean distance measure).

### 2.6. Assessment of the Hydrological Utility of SPPs

GR4J, a lumped model precipitation-runoff widely recommended for peak simulation, was utilized to simulate the hydrological flow in the study. The model requires daily areal precipitation in mm (P) and daily areal potential evapotranspiration in mm (E) as the input, while streamflow in mm per day (Q) is required to calibrate and validate the model. A description of the technical concepts of GR4J has been provided in the hydrological model section of the Supplementary Material above (Supplementary Text S4 and Table S2). Calibration and validation of the model are performed using HydroPSO, a multi-OS and model-independent package based on the Particle Swarm Optimization (PSO) technique [55]. PSO is a population-based stochastic optimization technique used to explore a delimited search space with a swarm of particles to find the best set of parameters required to maximize (or minimize) a user-defined objective function. The model was calibrated and validated based on the NSE value, chiefly due to its sensitivity to extreme flow NSE is more sensitive to extreme flow than RMSE, or Kling–Gupta Efficiency (KGE) [56,57]. Furthermore, performance indicators, namely, KGE, RMSE, and PBIAS, were also calculated to evaluate the performance of the hydrological model. The calibration period was selected from 2003 to 2007, while the model was validated using other datasets from 2010 to 2014. The calibration and validation period were selected on the basis of the availability of discharge datasets. After calibration and validation using gauge observed rainfall, the GR4J model was used to simulate the hydrological flow from 2003 to 2014 at the outlet using the six best-ranked SPPs by keeping the same model's calibrated parameters. Due to the constraint in data availability, ten days were selected as a warm-up period in this study. The simulated streamflow using the GR4J model forced with various SPPs was compared with ground observed discharge to assess the hydrological utility of SPPs. Four widely used performance indicators for hydrological simulations, NSE, KGE, RMSE, and PBIAS, were employed in this study for both the daily and the monthly time scales.

## 3. Results

In this section, the results of different performance indicators matrices—such as POD, FAR, ACC, CSI, RMSE, CC, PBIAS, and MAE—are discussed first. This is followed by a ranking of SPPs at each station and hydrological simulation results using the selected SPPs.

### 3.1. Performance Comparison among SPPs against Rain Gauge Observation

3.1.1. Performance Comparison Regarding the Spatial Distribution

The section compares the POD and RMSE of SPPs across the basin, as shown in Figure 3. Spatial variation in other categorical (FAR, ACC, CSI) and continuous (CC, PBIAS, and MAE) variable performance indicators are available in Supplementary Figures S3–S8. SPPs performance in terms of categorical and continuous variable performance indicators across the basin was inconsistent, i.e., the values varied from station to station (refer to Supplementary Figures S3–S8). For POD, the best performance is found for IMERG_Early, IMERG_Late, and PERSSIAN_CDR (0.7–0.8) across the basin, followed by CMORPH, PERSIANN (0.6–0.7). In addition to the overall high POD values, CMORPH, IMERG_Early, IMERG_Late, PERSIANN_CCS, and PERSIANN_CDR all showed a minor POD variation across the basin (Figure 3). This indicates that these products have persistently high detectability with low spatial variance, which means these SPPs are SPPs are robust across complex topography. On the other hand, widely used SPPs—CHIRPS (0.2–0.6) and IMERG_Final (0.6–0.8)—demonstrate a significant variation in POD across the basin. Both CHIRPS and IMERG_Final are adjusted with monthly ground-based observation and gridded gauge-analysis products—Global Precipitation Climatology Centre (GPCC) dataset—respectively, at a global scale, which might have a negative impact on their detectability of light rain events by enhancing the magnitude of stronger rainfall over complex topography. A previous study by Jiang et al. [58] also found similar results in the complex topography of China, where unadjusted IMERG_Early and IMERG_Late showed a better capability of detecting rain than GPCC-adjusted IMERG_Final.

(**a**)

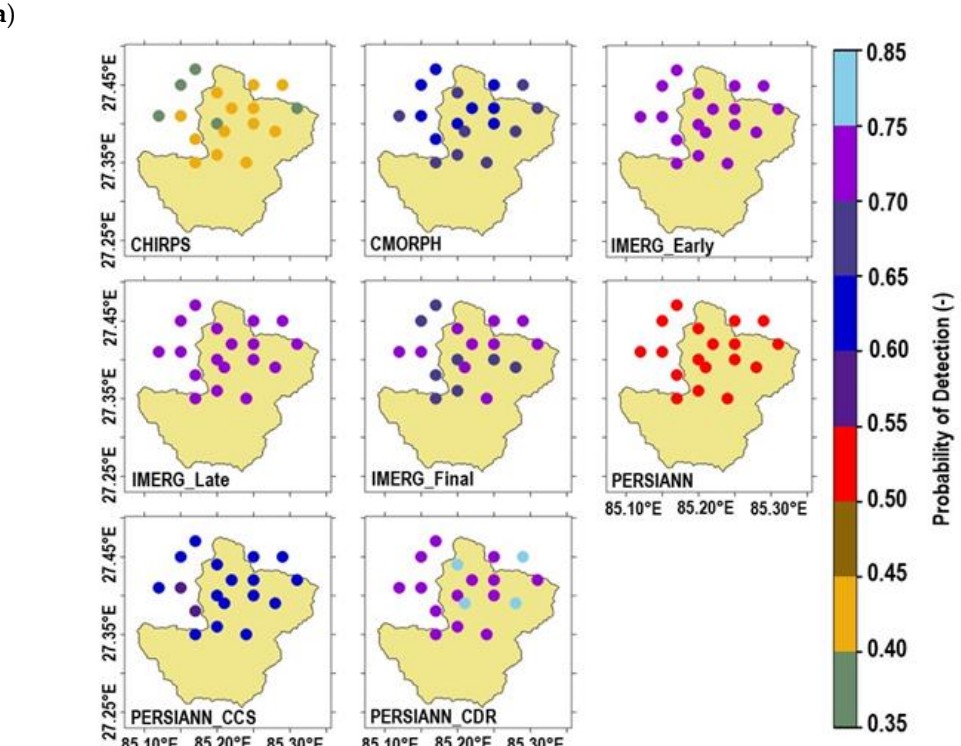

**Figure 3.** *Cont.*

(**b**)

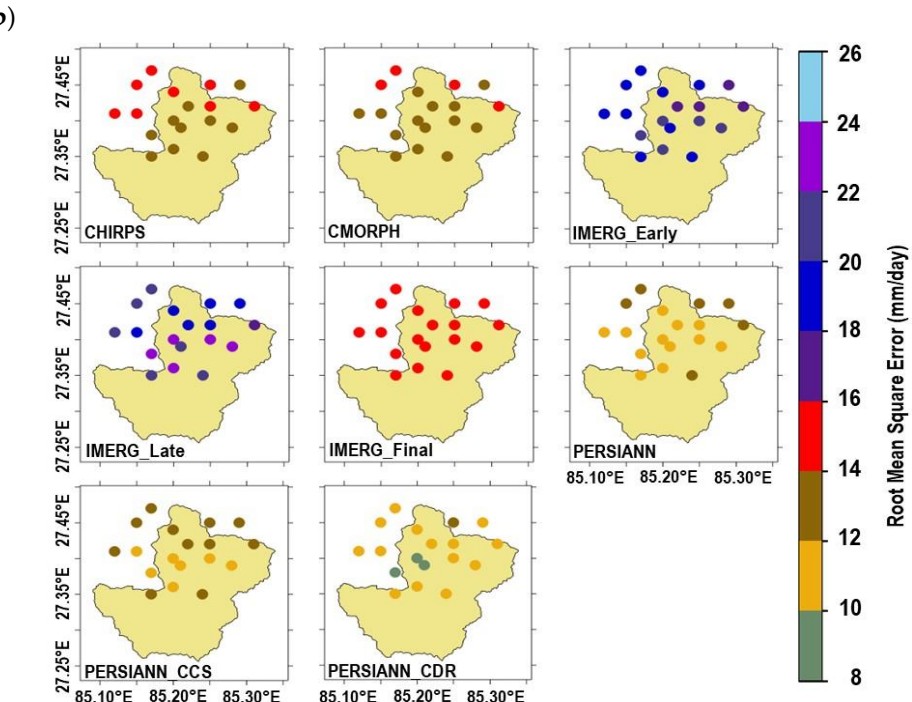

**Figure 3.** Spatial variation in the performance of all datasets in terms of POD (**a**) and RMSE (**b**) correspondence to each rain gauge station of Simat Khola River Basin from 2003 to 2018.

Despite the poor detectability, CHIRPS is competent in terms of matching the precipitation magnitude to the ground observation (Figure 3), which fortifies the above-mentioned speculation. Much like for POD, CMPROH, PERSIANN, and PERSIANN_CDR again stood out and showed similar performance across the basin in terms of RMSE (10 to 15 mm/day). PERSIANN_CDR showed the lowest RMSE at three stations near the southwest of the basin. On the other hand, IMERG_Final outperformed IMERG_Late and IMERG_Final. For IMERG_Late, the station located in the lower part of the basin has a higher RMSE than the one in the upper end. Despite good detectability of rainfall events, the worst performance in terms of RMSE was observed in IMERG_Early and IMERG_Late. Overall, IMERG_Final outperforms the two other IMERG products, both in terms of POD and RMSEs. The CC, PBIAS, and MAE correspondence to each station was observed as 0–0.4, −50–100%, and 4–10 mm, respectively, for all SPPs (Supplementary Figures S6–S8). The precipitation estimates from CHIRPS, CMORPH, IMERG_Late, IMERG_Final, and IMERG_Early, are better correlated (correlation coefficient range: 0.2 to 0.4) with ground-based observed precipitation as compared to the other SPPs (See Supplementary Figures S6–S8).

3.1.2. The Lumped Performance Comparison

Figure 4 lumps together all the stations and separately examines categorical (POD, FAR, ACC, and CSI) and continuous (RMSE, CC, PBIAS, and MAE) variable performance indicators. Overall, in terms of the ability to detect precipitation event-based indicators (POD, FAR, ACC, and CSI), CHIRPS performed the most poorly, while CMORPH, IMERG_Early, IMERG_Final, IMERG_Late, and PERSIANN_CDR performed similarly well. There was a high POD range (0.65–0.85) for CMORPH, IMERG_Early, IMERG_Late, IMERG_Final, and PERSIANN_CDR and a low POD range (0.40–0.45) for CHIRPS. FAR was observed to be 0.35–0.55, with extreme values corresponding to a few stations in PERSIANN_CCS and PERSIANN_CDR, where the probability of false alarm of the precipitation events exceeded more than 0.55. CMORPH, IMERG_Early, IMERG_Late, and IMERG_Final showed high ACC values (more than 0.76). The CSI of all SPPs, excluding CHIRPS, PERSIANN_CCS, and PERSIANN was in the range of 0.40–0.60.

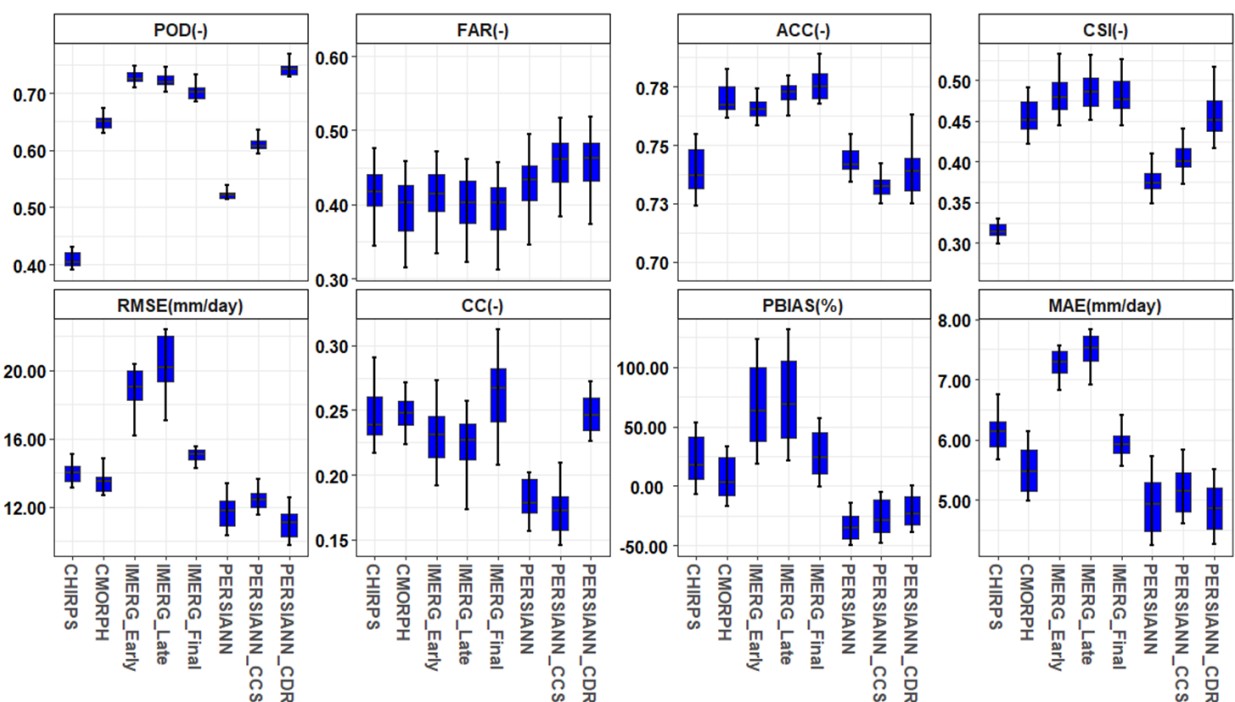

**Figure 4.** Box plot of categorical (POD, FAR, ACC, CSI) and continuous (RMSE, CC, PBIAS MAE) variable performance indicators at each station across the Simat Khola River Basin from 2003 to 2018.

In terms of the overall continuous variable performance, PERSIANN_CDR, CMORPH, and IMERG_Final surpassed the other SPPs across the basin. IMERG_Early and IMERG_Late displayed a high RMSE of more than 16 mm, approximately, while PERSIANN_CDR presented a low RMSE of almost less than 12 mm for most stations. The performance of all SPPs in MAE was comparable with RMSE because RMSE and MSE are highly correlated. However, despite enhanced performance in terms of RMSE and MAE, PERSIANN_CCS and PERSIANN showed the lowest correlation with gauge-based observed rainfall (CC less than 0.20). On the other hand, PERSIANN_CDR and IMERG_Final outperformed in the case of CC on the daily time scale, followed by CMORPH and CHIRPS. In PBIAS, all SPPs overestimated precipitation except for PERSIANN, PERSIANN_CDR, and PERSIANN_CSS. On balance, IMERG_Final showed a comparatively better performance than IMERG_Late and IMERG_Early in continuous variable performance indicators while performing more poorly in detecting rainfall events. IMERG_Final is an adjusted dataset with a GPCC precipitation on-time monthly scale that can effectively minimize the systematic satellite errors, particularly to a grid that lies near the GPCC station only on a monthly time scale. However, it can also increase additional uncertainty related to other grid cells not located near GPCC stations and having lower temporal resolution than a month [59]. These results also correspond to previous studies, e.g., Jiang et al. [58], Chen et al. [59], and Wang et al. [26].

### 3.2. Rank of SPPs

After determining the eight performance indicators for the SPPs, a payoff matrix was developed using all the indicators at each station. The weightage of these indicators was assigned by using the entropy method. Lastly, the Lp of SPPs was determined at each station using compromise programming. SPPs were ranked based on Lp values; the lowest value represents the best-performing product at that point. The calculated Lp values corresponding to the total 18-gauge stations of the Simat Khola River Basin were observed to be inconsistent (Supplementary Table S3). However, CMPROH and IMERG_Final received the lowest values at various stations. The station-wise ranking has been provided in Supplementary Table S4.

Furthermore, the mean of Lp was employed to rank SPPs on the basin scale. CMORPH and IMERG_Final ranked first and second, respectively, while PERSIANN_CCS and IMERG_Early ranked last and second-last, respectively, based on their ability to represent precipitation (refer to Table 2). These results contradict a previous study by Kumar et al. [39], which found that CMOPRH significantly underestimated the rainfall and performed worse than TRMM 3B42, a previous version of IMERG_Final, over the Gandaki River Basin Nepal. However, CMORPH has consistently performed well in all performance indicators across the selected river basin, per our results. Our study differs from the erstwhile study in terms of selected performance indicators, temporal evaluation scale, study areas, and rain gauge network density, which may explain the contrasting results. Our study used a small region with a high rainfall network density (23.13 km$^2$/station) and employed indifferent performance indicators daily to rank SPPs. However, Kumar et al. [39] worked with comparative large study areas with a low rainfall network density (7717 km$^2$/station) to evaluate the SPPs on a monthly time scale. PERSIANN_CCS and IMERG_Early are the lowest-performing SPPs because both are entirely satellite products; they significantly underestimated the rainfall amount in the monsoon season ( Supplementary Figure S2). Previous studies, e.g., Chen et al. [59] and Beck et al. [25], also found a significant underestimation of rainfall, particularly for PERSIANN_CCS. In addition, our results are consistent with earlier studies (e.g., Tan et al. [7]; Beck et al. [25]; Lu et al. [60]). The six best-ranked SPPs were selected to simulate the discharge at the SKRB outlet, with the assumption that the last two ranked SPPs might not suit the hydrological application.

**Table 2.** The assigned rank of SPPs based on categorical (POD, FAR, ACC, and CSI) and continuous (RSME, CC, PBIAS, and MAE) variables performance indicators on a daily time scale by employed compromise programming MCDM for Simat Khola River Basin, Nepal, as a representative region of morphological complex Himalayan Region.

| Dataset | Average LP | Rank |
|---|---|---|
| CMORPH | 0.045665 | 1 |
| IMERG_Final | 0.091280 | 2 |
| PERSIANN_CDR | 0.117600 | 3 |
| PERSIANN_CCS | 0.202612 | 4 |
| PERSIANN | 0.215901 | 5 |
| IMERG_Early | 0.218660 | 6 |
| CHIRPS | 0.231086 | 7 |
| IMERG_Late | 0.242115 | 8 |

*3.3. Suitability of SPPs for Streamflow Estimation*

3.3.1. Calibration Results of GR4J Model

The GR4J model development process first involved calibration from 2003 to 2007; in the second step, the model was validated from 2010 and 2014 using different gauge observed datasets. Four parameters, the maximum capacity of production storage (X1), Groundwater exchange coefficient (X2), One day ahead maximum capacity of routing store (X3), and time base of unit hydrograph UH1 (X4), were adjusted to find the best simulation of discharge during the calibration process. The best simulation of runoff at the outlets of Simat Khola River was obtained for parameters X1 = 390.965 mm, X2 = 2.85085 mm, X3 = 20.050 mm, and X4 = 1.1 days. Furthermore, in the best simulation, NSE and KGE were 0.66 and 0.79, respectively, on a daily time scale. The monthly performance was improved; NSE and KGE of the model were 0.91, and 0.95 on a monthly time scale, respectively. Based on the NSE values, the model was classified as well-calibrated and best-calibrated at the daily and monthly time scales. Furthermore, the model recorded three peaks on a monthly time scale and overestimated the correspondence to the two remaining peaks during the calibration period; the time to peak, however, was captured completely. During the validation period, on a monthly scale, the model overestimated the peak twice and underestimated it thrice in the last three validation periods. The

model performance indicators, NSE and KGE, were almost equal to the calibration period's performance indicators on both time scales during the validation period. NSE and KGE at daily and monthly time scales were 0.61–0.70 and 0.85–0.90, respectively. The model failed to capture the discharge peaks in the last two years of the validation period; besides this, the model captured the timing of the peak entirely on the monthly time scale. The calibration and validation of the GR4J model at the Simat Khola River Basin outlet are illustrated in Figure 5, respectively.

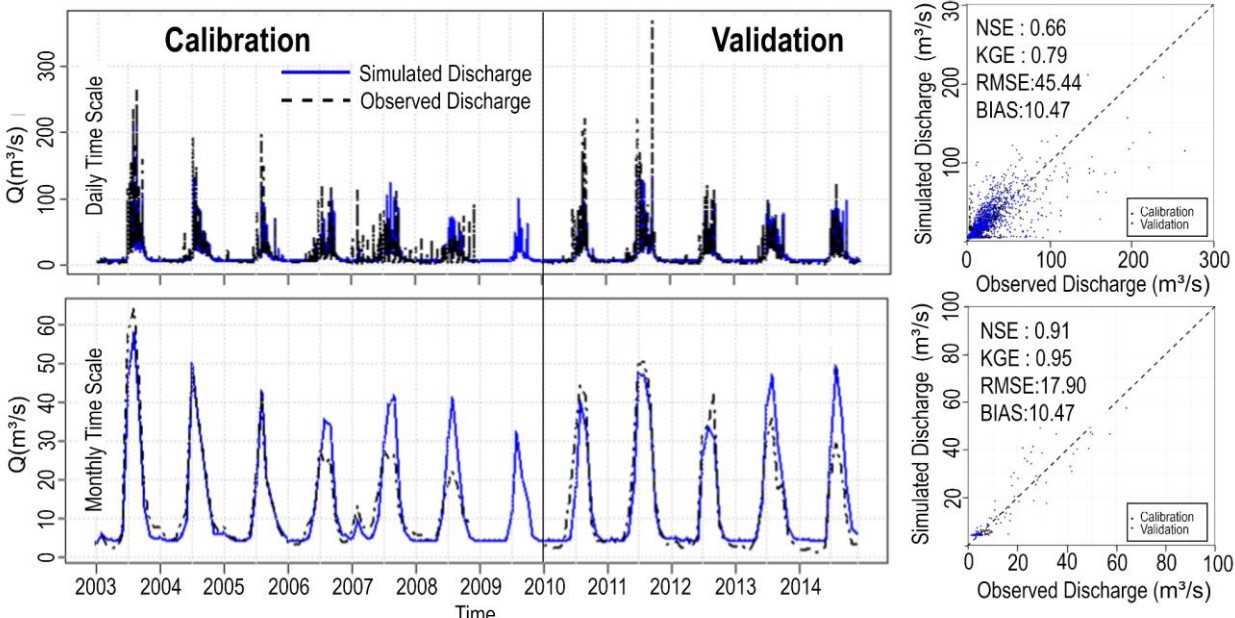

**Figure 5.** Time series and scatter plot of GR4J model-based simulated and gauge-based observed discharge at the outlet of Simat Khola River Basin during a calibration (2003 to 2007) and validation period (2010 to 2014) daily and monthly basis along with model performance indicators.

### 3.3.2. Discharge Simulation Capability of SPPs

The calibrated and validated model was used to simulate the runoff at the basin outlet from 2003 to 2014 using the selected SPPs—CHIRPS, CMORPH, IMERG_Late, IMERG_Final, PERSIANN, PERSIANN_CDR—from the previous sections. The NSE values for CHIRPS, IMERG Late, and IMERG Final were all negative; the remaining SPPs had positive NSE but a lower-than-acceptable limit [61]. Comparatively, the higher NSE value was observed in the PERSIANN_CDR dataset on the daily time scale. Since NSE is regarded as a reliable indicator for capturing severe discharge events, it is clear from the simulation results that no products were acceptable for hydrological extreme event modeling on a daily time scale (refer to Figure 6a). Furthermore, even though IMEGR_Final and CHIRPS showed a negative range of NSE (unacceptable in any case), both SPPs showed better performance in terms of KGE and BIAS than PERSINANN, which showed a higher NSE value.

With the exception of PERSIANN and IMERG Late, all precipitation products displayed the expected range of NSE and KGE at a monthly period. Although the performance of these SPPs, PERSIANN and IMERG Late, was unacceptable, their capability to simulate on a monthly basis was better than that of the daily scale and monthly simulation. It is interesting to note that on a monthly basis, the discharge simulation performance of CHIRPS and IMERG_Final significantly improved and reached an acceptable range from unacceptable daily simulation results. In addition, CMORPH displayed the best NSE values on a monthly time scale (refer to Figure 6b). Except for PERSIANN and PERSIANN_CDR, all SPPs captured the peak timing and magnitude on a monthly time scale (Supplementary Figures S9 and S10). In addition, Figure 6a,b illustrate the explicit representation of simulated discharge using all SSPs—CHIRPS, CMORPH, IMERG_Late, IMERG_Final,

PERSIANN, and PERSIANN_CDR—for the selected periods of study along with model performance indicators on the daily and monthly time scale.

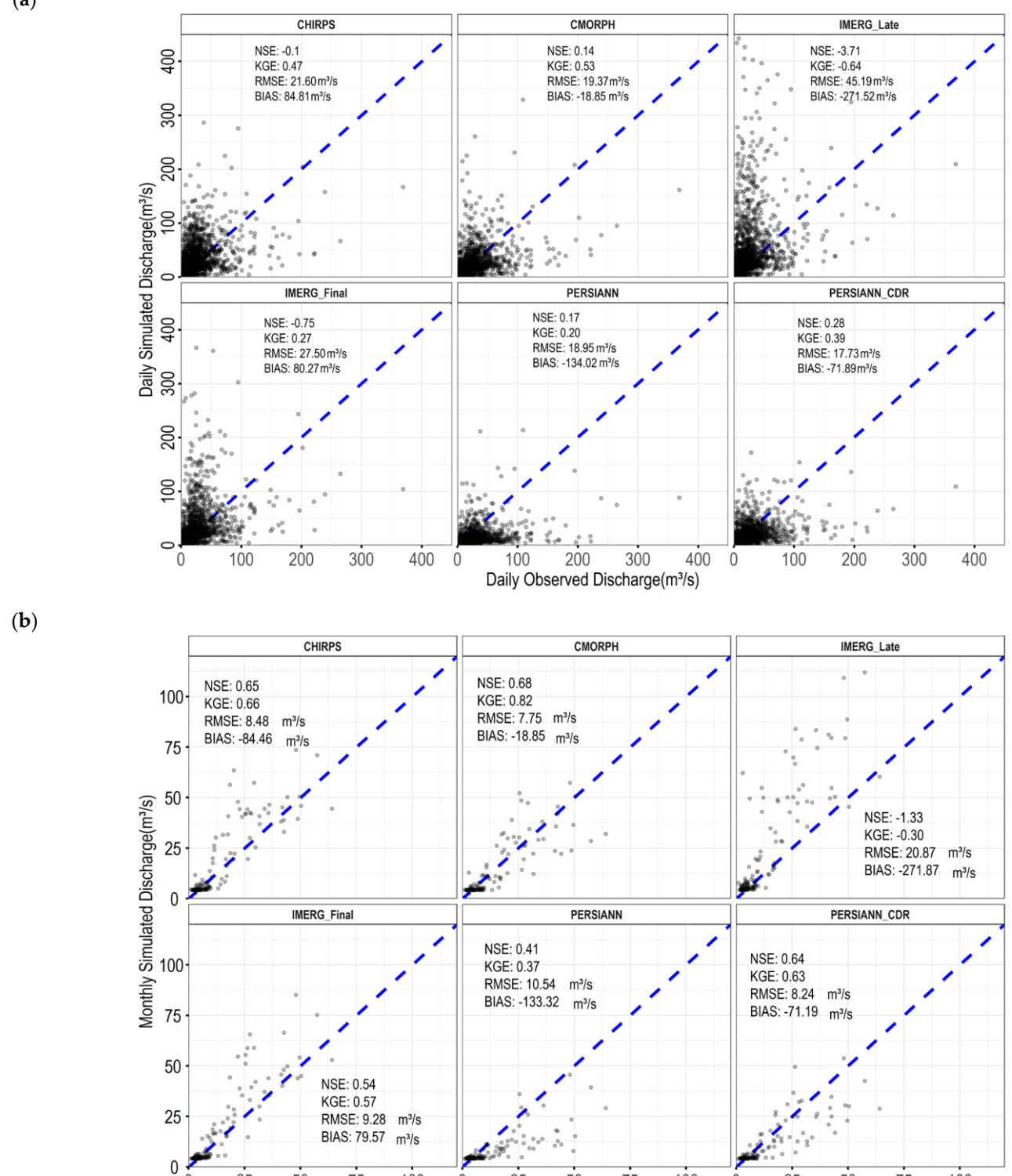

**Figure 6.** Scatter plots of daily (**a**) and monthly (**b**) simulated streamflow from CHIRPS, CMORPH, IMERG_Late, IMERG_Final, PERSIANN, and PERSIANN_CDR against corresponding gauge-based observed streamflow at the main outlet of the basin along with hydrological performance indicators NSE, KGE, RMSE, and BIAS. The blue dotted line denotes a 1:1 line.

## 4. Discussion

Continuous improvements in global precipitation datasets allow researchers and users to utilize them as competent alternatives to gauge observations, especially over data-sparse regions and ungauged river basins. However, due to the uncertainties inherent with SPPs, their efficacy differs from basin to basin. It is often recommended to evaluate all the precipitation products before application. Therefore, this study attempted to evaluate the prevailing SPPs featuring adequate spatiotemporal resolutions to represent the precipitation characteristics at the basin scale, namely, CHIRPS, CMORPH, IMERG_Early, IMERG_Final, IMERG_Late, PERSIANN, PERSIANN_CCS, and PERSIANN_CDR for Simat Khola River Basin in Nepal, as a representative region of the morphological complex Himalayan region. On a daily basis, SPPs were compared to gauge-based observed precipitation, while on both the daily and monthly scales, the simulated streamflow using the GR4J model forced with SPPs, and ground observed streamflow were compared. Altogether, all selected SPPs performed differently at different rain gauge stations across the basin.

On the daily time scale, CMORPH, IMERG_Early, IMERG_Late, IMERG_Final, and PERSIANN_CDR consistently outperformed the majority of the indicators. Furthermore, CHIRPS, CMORPH, and PERSIANN_CDR demonstrated a high correlation with observed rain gauge data at each station. The performance of all datasets based on the selected indicators was ambiguous. There were no SPPs that performed optimally across all indicators. For instance, our research showed that CHIRPS had the best PBIAS performance but performed the worst in terms of POD. Several previous studies worldwide also support our results. For instance, Le et al. [28] found the performance of CHIRPS in terms of POD (<50) was the worst, but in terms of RMSE, it was best (13 mm/day) during both periods, wet and dry, across various basins in Vietnam. Dandridge et al. [18] results indicate better performance for CHIRPS in RMSE but worse in terms of Bias in the Lower Mekong Delta. Therefore, selecting the best SPP to use as an alternative to ground observed precipitation remains difficult.

To nullify this emerging use, for the first time, the ranking of SPPs for the Simat Khola River Basin has been conducted using compromise programming in the second part of the study. This provided a clear understanding of the performance of all datasets and allowed the best-performing datasets to be chosen for hydrological simulations. The rank assigned to SPPs based on the ability to represent precipitation processes at the basin was arranged in descending order, CMORPH, IMERG_Final, PERSIANN_CDR, PERSIANN, CHIRPS, IMERG_Early, IMERG_Late, and PERSIANN_CCS. The ranking of SPPs varied from station to station. This could be the result of various factors such as the location of the most precipitating stations, which have biases between the observed precipitation as well as the SPPs. Many of Nepal's precipitation stations are located within the lower elevation zone, which may not capture the actual amount of daily precipitation [62–64].

In the final part of this study, the evaluation of selected SPPs was performed based on the ability to capture streamflow at basin outlets by simulating the discharge using the GR4J precipitation-runoff model. As per daily simulation results, IMEGR_Final and CHIRPS showed a negative range of NSE, which is inconsistent with previous studies by Yuan et al. [27] and Le et al. [28]. Their research indicated that IMERG_Final moderately acceptable performance based on NSE (>0.50) at a daily time scale across basins in Vietnam [18,27,28]. In the case of other SPPs, our results are comparable with the previous studies worldwide [27,28]. Overall, it can be concluded, based on the study, that all SPPs failed to simulate the discharge at the basin outlets on a daily time scale, which may be due to different causes. Snowmelt models were not included in the hydrological simulation due to inadequacy and insufficiency in the observed datasets; therefore, one reason could be an accumulation of snowmelt in the discharge data.

For the monthly time-scale, on the other hand, the GRJ4 model forced with SPPs was found to reproduce streamflow within the acceptable range. In both study periods, discharge simulation performed better on a monthly scale than on a daily scale. With the exception of PERSIANN and IMERG Late, all SPPs displayed the expected range of NSE and KGE on a

monthly time-scale. Although the performance of these SPPs, PERSIANN and IMERG Late, was inadequate, their capacity to simulate on a monthly basis was better than daily basis, and the results of the monthly simulation are consistent with the previous study [26–29,60]. The main outcome of their research shows that IMERG_Final was in the common list of the SSPs which have demonstrated the best hydrological performance among the other SSPs. The research by Yuan et al. [27] in the same region of southeast Asia only compared three SSPs, and they have acceptable hydrological capabilities of IMERG_Final were better than other IMERG products. Moreover, research by Dandridge et al. [18], Yuan et al. [27], Dhanesh et al. [29], and others in this region have not included the CMORPH product in their evaluation, which is one of the best performing products in our basin.

The bias correction and snow accumulation model have not been included in this paper. Therefore, it is highly recommended that other researchers extend this work by using a bias correction algorithm, considering a snowmelt model, and analyzing the dataset's performance after bias correction. Nevertheless, based on this study, CMORPH is the best-recommended product for application as an alternative to observed precipitation data, filling of missing observed data, and precipitation-runoff modeling on the monthly time scale in the study basin. These findings can benefit researchers in the selection of reliable SPPs as an alternate source of rain gauge in the study basin in Nepal and the larger data-sparse Himalayan region. The study's recommendation will also be valuable in guiding policymakers and engineers in determining the appropriate data for flood prediction, forecasting, and early warning to mitigate flood risks as well as the associated impact on the local communities. Furthermore, the data evaluation information can assist algorithm developers in correcting errors and improving the selected SPPs.

## 5. Conclusions

The Asian summer monsoon periodically brings excessive precipitation to the Himalayan basin, which causes life-threatening hazards of floods and landslides in tandem with enormous economic losses. The early warning and monitoring of such hazards require the accurate representation of the areal precipitation, which is challenging to procure from the sparsely distributed rain gauge observations. Meanwhile, the regional hydro-meteorological modeling also requires the reliable input of precipitation fields. This study aims to select the best SPP over the Simat Khola River Basin for local rain gauge observations. Firstly, the SPPs were evaluated based on categorical and continuous variable certification. Second, compromise programming was applied to rank SPPs for the basin, facilitating a clear understanding of the performance of all datasets and the selection of the best-performing datasets for hydrological simulation. Finally, the evaluation of selected SPPs was undertaken based on the ability to capture streamflow at the basin outlets by simulating the discharge using the GR4J precipitation-runoff model.

Altogether, every SPP performed differently at different rain gauge stations across the basin. Furthermore, CMORPH, IMERG_Final, and PERSIANN_CDR consistently stood out in terms of POD, FAR, ACC, CSI, RMSE, CC, PBIAS, and MAE daily. The rank assigned to SPPs based on the ability to represent precipitation processes at the basin in descending order is as follows: CMORPH, IMERG_Final, PERSIANN_CDR, PERSIANN, CHIRPS, IMERG_Early, IMERG_Late, and PERSIANN_CCS. On a daily scale, all precipitation products failed to accurately replicate the discharge at the basin's outlet. However, a total of three SPPs, i.e., CMORPH, IMERG_Final, and CHIRPS, were found to be within the acceptable range in their ability to simulate discharge on the monthly time scale. In outline, CMORPH and IMERG_Final are the most suitable SPPs to be used in the Simat Khola River Basin, Nepal, and other high-altitude basins as alternatives to ground-based precipitation. For future research, a few the current study's limitations need to be taken into account.

- Our analysis is restricted to a small part of the Himalayan region, and it might not be robust for other morphologically complicated mountains. As a result, further research should be conducted on a wide range of small to large watersheds using our novel comprehensive framework, which can systematically rank different datasets

and identify the most suitable for hydro-meteorological application in the ungauged river basin.

- Since no SPPs were found to be outstanding (like previous studies) in capturing rainfall events and magnitude and simulating the discharge, future research can therefore be focused on bias correction of SPPs to improve the performance of SPPs.
- A snowmelt model has not been taken into account in this work while simulating discharge using a hydrological model. In addition, our hydrological model has not been calibrated and validated using each SPPs incorporating snowmelt model that might be included in future research to investigate the precipitations' product-specific simulation capability.
- This study has only evaluated the SSPs for daily and monthly streamflow simulation, but in terms of hydrological studies, the time of occurrence of any extrema event is critical; therefore, further investigation should apply at sub-daily time scales.

**Supplementary Materials:** The following supporting information can be downloaded at: https://www.mdpi.com/article/10.3390/rs14194810/s1 [23,36–38,40,42–45,65–67], Figure S1: Time series and box plots of precipitation (2003–2018) daily monthly, and annually time scale of Simat Khola River Basin, Nepal; Figure S2: Schematic model structure of GR4J with GR4JSG; Figure S3: Spatial variation in the performance of all datasets in terms of FAR correspondence to each rain gauge station of Simat Khola River Basin from 2003 to 2018; Figure S4: Spatial variation in the performance of all datasets in terms of ACC correspondence to each rain gauge station of Simat Khola River Basin from 2003 to 2018; Figure S5: Spatial variation in the performance of all datasets in terms of CSI correspondence to each rain gauge station of Simat Khola River Basin from 2003 to 2018; Figure S6: Spatial variation in the performance of all datasets in terms of CC correspondence to each rain gauge station of Simat Khola River Basin from 2003 to 2018; Figure S7: Spatial variation in the performance of all datasets in terms of PBIAS correspondence to each rain gauge station of Simat Khola River Basin from 2003 to 2018; Figure S8: Spatial variation in the performance of all datasets in terms of MAE correspondence to each rain gauge station of Simat Khola River Basin from 2003 to 2018; Figure S9: Time series plot of daily observed and simulated discharge using GR4J model forced with selected SPPs—Observed (a), CHIRPS (b), CMORPH (c), IMERG_Final (d), PERSIANN (e), and PERSIANN_CDR (f) at the outlet of Simat Khola River Basin (2003–2014); Figure S10: Time series plot of monthly observed and simulated discharge using GR4J model forced with selected SPPs—Observed (a), CHIRPS (b), CMORPH (c), IMERG_Final (d), PERSIANN (e), and PERSIANN_CDR (f) at the outlet of Simat Khola River Basin (2003–2014); Table S1: Sources and descriptions of the datasets used in the present study; Table S2: Summary of different parameters of GR4J model and its acceptable range; Table S3: Calculated Lp parameter using Compromise Programming and an overall rank of SPPs for the selected river basin; Table S4: The calculated rank of SPPs for the selected river basin.

**Author Contributions:** S.K. conceived the study, carried out the data analysis and contributed to the development of the code. S.K., E.P., and T.B. reviewed the mathematical framework. S.K. prepared the first draft with the help of E.P. and T.B., D.B. collected the data. G.A., S.G., E.P., T.B., M.P., and J.W. contributed to the discussion and interpretation of the results. E.P. acquired funding. All authors have read and agreed to the published version of the manuscript.

**Funding:** This research was funded by the CGIAR (Consultative Group of International Agricultural Research) Program (CRP) on Water, Land, and Ecosystems (WLE) and Ministry of Education of Singapore (#Tier2 MOE-T2EP402A20-0001).

**Data Availability Statement:** The datasets generated during and/or analyzed during the current study are available from the corresponding author on reasonable request.

**Acknowledgments:** This research was funded by the CGIAR (Consultative Group of International Agricultural Research) Program (CRP) on Water, Land, and Ecosystems (WLE). We also acknowledge DHM, Nepal for providing the ground-based observed data. This research was supported by the Earth Observatory of Singapore via its funding from the National Research Foundation Singapore and the Singapore Ministry of Education under the Research Centres of Excellence initiative. This work comprises EOS contribution number 477.

**Conflicts of Interest:** The authors declare no conflict of interest.

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
