# Peer review of "Assessing the Performance of the Satellite-Based Precipitation Products (SPP) in the Data-Sparse Himalayan Terrain"

_remotesensing, doi:10.3390/rs14194810_

Round 1
Reviewer 1 Report
Assessing the performance of the satellite-based precipitation products (SPP) in the data-sparse Himalayan terrain:
· In the last paragraph of the Introduction, the authors should mention the weak point of former works (identification of the gaps) and describe the novelties of the current investigation to justify that the paper deserves to be published in this journal.
· Lines 37-38, cite these recent useful papers on the importance of rain gauges to improve the literature and to show the importance of your work:
Evaluation of Satellite Precipitation Estimates Over Omo–Gibe River Basin in Ethiopia
Calibration of mass transfer-based models to predict reference crop evapotranspiration
· Discuss more the discharge simulation capability of SPPs.
· How can expand the results to other regions with similar/different climates?
· At the end of the manuscript, explain the implications and future works considering the outputs of the current study.
Author Response
Dear Reviewer,
Thank you very much for your valuable time in reviewing our manuscript. We have tried our level best to address the points raised, and the response letter is attached below for your kind reference.
Best Regards,
All authors of the Manuscript

Reviewer 2 Report
The present study compares satellite-based precipitation products (SPPs) with rain gauges over Nepal and uses them as forcing inputs to simulation hydrological models. It provides additional information on SPPs performances at a relatively new site rather than in a more intense research area such as China, India, South America, or CONUS. I recommend a major revision before consideration for publication.
The study only investigated a small watershed (~300 km²), which may affect their interpretation results in general. At least two more small watersheds should be included to draw a valid general conclusion.
For SPPs that can provide sub-daily precipitation datasets, please clarify the data pre-processing steps. How do the authors aggregate these datasets from a sub-daily to a daily basis? In Nepal, how is a precipitation day defined ̣(from what time to what time) ?
In your discussion of SPP performances, please include the following references.
Yuan, F., Zhang, L., Soe, K. M. W., Ren, L., Zhao, C., Zhu, Y., ... & Liu, Y. (2019). Applications of TRMM-and GPM-era multiple-satellite precipitation products for flood simulations at sub-daily scales in a sparsely gauged watershed in Myanmar. Remote Sensing, 11(2), 140.
Satgé, F., Defrance, D., Sultan, B., Bonnet, M. P., Seyler, F., Rouche, N., ... & Paturel, J. E. (2020). Evaluation of 23 gridded precipitation datasets across West Africa. Journal of Hydrology, 581, 124412.
Dandridge, C., Lakshmi, V., Bolten, J., & Srinivasan, R. (2019). Evaluation of satellite-based rainfall estimates in the lower mekong river basin (southeast asia). Remote Sensing, 11(22), 2709.
Le, M. H., Lakshmi, V., Bolten, J., & Du Bui, D. (2020). Adequacy of satellite-derived precipitation estimate for hydrological modeling in Vietnam basins. Journal of Hydrology, 586, 124820.
Dhanesh, Y., Bindhu, V. M., Senent-Aparicio, J., Brighenti, T. M., Ayana, E., Smitha, P. S., ... & Srinivasan, R. (2020). A comparative evaluation of the performance of CHIRPS and CFSR data for different climate zones using the SWAT model. Remote Sensing, 12(18), 3088.
Some other comments:
Table 1:
Latency for rain gauge - 1y&5y: Please clarification
Please correct latency for IMERG Early, IMERG Late, and IMERG Final
Please replace reference [38] with the latest reference for IMERG product as follows:
Huffman, G. J., Bolvin, D. T., Braithwaite, D., Hsu, K. L., Joyce, R. J., Kidd, C., ... & Xie, P. (2020). Integrated multi-satellite retrievals for the global precipitation measurement (GPM) mission (IMERG). Satellite precipitation measurement, 343-353.
L294 SPP are immune from the complex topography -> SPPs are robust …
Figure 3 – The current color theme did not show a variation in POD or RMSE. The range of POD or RMSE should be narrowed.
L340 PERCIAN_CDR -> PERSIANN_CDR
L379 Our study used small regions -> Our study focused on a small region [You only use one watershed but if you include more than one watershed, a plural noun can be used here]
L379-L386: Please provide the rainfall network in your study in km²/station and Kumar et al in km²/station
Author Response

(The authors gave the same response as above.)

Reviewer 3 Report
This study evaluates the performance of Satellite-based Precipitation Products (SPPs) in the Himalayas region by comparing different datasets, using performance indicators and Multi-Criteria Decision-Making. It is a suitable topic for Remote Sensing MDPI journal. The manuscript is professionally written, clear, and easy to read. Therefore, I would suggest it to be accepted as it is.
Author Response

(The authors gave the same response as above.)

Round 2
Reviewer 1 Report
I appreciate the authors addressing the comments. The manuscript can be accepted in its current form. Congrats!
Reviewer 2 Report
I happy that the authors address all my concerns.